# Translation and cultural adaptation of drug use stigma and HIV stigma measures among people who use drugs in Tanzania

Linda B. Mlunde[1]*, Lisa R. Hirschhorn[2], Laura Nyblade[3], Nan E. Rothrock[2], Erasto V. Mbugi[4], Judith T. Moskowitz[2], Sylvia Kaaya[5], Claudia Hawkins[6], Germana Leyna[7], Jessie K. Mbwambo[8]

1 Department of Community Health, School of Public Health and Social Sciences, Muhimbili University of Health and Allied Sciences, Dar es Salaam, Tanzania, 2 Department of Medical Social Sciences, Feinberg School of Medicine, Northwestern University, Chicago, Illinois, United States of America, 3 Health Practice, RTI, International, Washington, DC, United States of America, 4 Department of Biochemistry, School of Medicine, Muhimbili University of Health and Allied Sciences, Dar es Salaam, Tanzania, 5 Department of Psychiatry and Mental Health, School of Medicine, Muhimbili University of Health and Allied Sciences, Dar es Salaam, Tanzania, 6 Department of Medicine, Feinberg School of Medicine Northwestern University, Chicago, Illinois, United States of America, 7 Department of Epidemiology and Biostatistics, School of Public Health and Social Sciences, Muhimbili University of Health and Allied Sciences, Dar es Salaam, Tanzania, 8 Department of Psychiatry and Mental Health, Muhimbili National Hospital, Dar es Salaam, Tanzania

* lindasozy@gmail.com

## Abstract

### Introduction

People who use drugs (PWUD) experience stigma from multiple sources due to their drug use. HIV seroprevalence for PWUD in Tanzania is estimated to range from 18 to 25%. So, many PWUD will also experience HIV stigma. Both HIV and drug use stigma have negative health and social outcomes, it is therefore important to measure their magnitude and impact. However, no contextually and linguistically adapted measures are available to assess either HIV or drug use stigma among PWUD in Tanzania. In response, we translated and culturally adapted HIV and drug use stigma measures among Tanzanian PWUD and described that process in this study.

### Methods

This was a cross-sectional study. We translated and adapted existing validated stigma measures by following a modified version of Wild's ten steps for translation and adaptation. We also added new items on stigmatizing actions that were not included in the original measures. Following translation and back translation, we conducted 40 cognitive debriefs among 19 PWUD living with and 21 PWUD not living with HIV in Dar es Salaam to assess comprehension of the original and new items. For challenging items, we made adaptations and repeated cognitive debriefs among ten new PWUD participants where half of them were living with HIV.

Research and Ethics Committee of Tanzania. Data requests can be directed to drp@muhas.ac.tz.

**Funding:** The study was supported by the Fogarty International Center of the National Institutes of Health (NIH) under Award Number D43TW010946. The research is solely the responsibility of the authors, and the NIH had no role in study design, data collection and analysis, decision to publish, or preparation of the manuscript. No any, among authors have received salary from NIH.

**Competing interests:** The authors have declared that no competing interests exist.

## Results

Most of the original items (42/54, 78%), response options and all items with new 12 stigmatizing actions were understood by participants. Challenges included response options for a few items; translation to Swahili; and differences in participants' interpretation of Swahili words. We made changes to these items and the final versions were understood by PWUD participants.

## Conclusion

Drug use and HIV stigma measures can successfully be translated and culturally adapted among Tanzanian PWUD living with and without HIV. We are currently conducting research to determine the stigma measures' psychometric properties and we will report the results separately.

## Introduction

Drug use continues to bring a tremendous burden to the global community. About 296 million people reported using drugs in 2021 [1] in the world. The most recent data available (2014), showed that there are an estimated 300,000 people who use drugs (PWUD) in Tanzania, with the Dar es Salaam region estimated to have the highest prevalence in the country [2]. PWUD are more likely to engage in behaviors that put them at risk of contracting infectious diseases such as human immunodeficiency virus (HIV) [3]. HIV prevalence in the general population in Tanzania is 4.9%, [4] but higher among PWUD ranging from 18 to 25% [2].

Stigma is a social process where an individual or a group of people is excluded, rejected, blamed or devalued after being judged because of having or being assumed to have a certain identity or certain health condition [5, 6]. Types of stigma include experienced or enacted, perceived, anticipated, and internalized stigma, all of which can occur in PWUD. Perceived stigma occurs when individuals feel that others are displaying stigmatizing attitudes towards them such as feeling they are not good people or they are dangerous to the health of others [7]. Anticipated stigma refers to the degree to which an individual expects that s/he will experience prejudice and discrimination from others in the future [8]. Internalized or self-stigma involves feelings of hopelessness, worthlessness, shame, blame, and self-hatred [5]. In addition, PWUD may experience enacted stigma or discrimination where they experience unjust or prejudicial treatment by others [9]. PWUD who are also HIV positive may face both drug use and HIV stigma complicating their livelihood, health status, and engagement in care [10].

In Tanzania, PWUD have reported experiencing stigma in health care settings and in the community [11]. Stigma results in poor health seeking behaviors and access to care [11, 12] which could further increase the negative effects of the condition. The stigma PWUD face can be associated with negative health and social outcomes. Stigma may lead individuals to socially withdraw and isolate themselves from relationships and activities [5]. It can also lead to loss of job opportunities and family rejection, as well as reluctance to access health services, including harm reduction services and for people living with HIV (PLHIV), HIV care and treatment [13–15].

These negative impacts of stigma among PWUD, including those living with HIV, highlight the need to design interventions to address stigma and the effects of stigma on health status and care seeking. It is necessary to measure stigma that PWUD face to understand its burden, design stigma-reduction interventions, and assess the impact of such interventions. Stigma

can be assessed by self-report using a patient-reported outcome measure (PROM). A number of HIV stigma [8, 16, 17] and drug use stigma PROMs [18–20] have been used in different contexts. However, to our knowledge, none of these existing PROMs have been validated in countries in Africa or adapted for use in Swahili (the common language used across ethnic groups in Tanzania) for use with PWUD.

To effectively use stigma measures in a new cultural and linguistic context, they have to be appropriately adapted for that target population [21]. This process includes both linguistic translation and cultural adaptation to ensure that the PROM is measuring the locally relevant constructs and experiences. Cognitive debriefing is an important step in cultural adaptation, designed to determine the understanding of a measure's questions and responses by individuals in a new setting. Cognitive debriefing includes testing questions and determining how they should be modified to make them more understandable to study participants and to also maintain the targeted areas of measurement [22]. In this article we describe the translation and cultural adaptation of validated drug use stigma and HIV stigma measures for use among Tanzanian PWUD including those living with HIV. This study does not include a psychometric evaluation of the measures which is being currently conducted and findings will be reported separately.

## Methods

### Study area

We conducted the study in Dar es Salaam, Tanzania. Tanzania has a population of 61.1 million; Dar es Salaam, the largest city in Tanzania, hosts 5.3 million people [23]. We conducted our study in three (Ilala, Kinondoni and Temeke) of the five districts in the city where a high number of PWUD congregate.

### Study population

We included PWUD living with HIV and those without HIV or of unknown status. HIV status was self-reported. For all participants, the inclusion criteria were i) history of ever using illicit drugs by any method, ii) aged 18 years and above, iii) living in Dar es Salaam for at least three months. Individuals using medically assisted treatment (MAT) were eligible. Exclusion criterion was being unwilling or unable to respond to questions due to other medical conditions. Community-based organizations which link PWUD to MAT, assisted with the recruitment of participants by word-of-mouth from clinics and the community. All participants who were approached agreed to participate and did not know the researchers before data collection.

### Study design

This was a cross-sectional study. We modified Wild's ten steps for translating and adapting PROMs [22]. The ten steps include 1) Preparation; 2) Forward Translation; 3) Reconciliation; 4) Back Translation; 5) Back Translation Review; 6) Harmonization; 7) Cognitive Debriefing; 8) Review of Cognitive Debriefing Results and Finalization; 9) Proofreading; and 10) Final Report writing [22]. LBM and a researcher with expertise in stigma modified the mentioned ten steps by introducing a step after Harmonization—expert review and development of new items. We reviewed the original drug use stigma and HIV stigma measures to evaluate how well they measured all aspects of stigma experienced by PWUD in Tanzania, based on a qualitative formative assessment conducted on experiences of stigma among PWUD in Dar es Salaam, Tanzania by LN, LBM, and JKM (manuscript under peer review). The formative assessment included in-depth interviews of 18 PWUD and 14 HIV care and treatment clinic

(CTC) staff. The aim of the formative assessment was to understand the types of stigma and their manifestations that PWUD face. We then developed new items to reflect the stigmatizing actions missing from the original validated measures based on the Tanzanian context.

**Step 1: Preparation.** Before the translation and cultural adaptation of the original measures to the Tanzanian context, we obtained permission from the authors. Adaptation may include modifications of the measures based on the context of use.

*Measures.* We selected measures which contained a mix of items assessing experienced, anticipated, perceived and internalized stigma (Table 1). The HIV stigma measures included: 1) the Abridged Berger Scale [24] assessing personalized stigma, negative self-image, public attitudes, and disclosure concerns; 2) the Internalized AIDS-Related Stigma Scale (IA-RSS) [16] measuring internalized stigma; and 3) the HIV Stigma Mechanism Measure which measures internalized stigma, anticipated stigma and enacted stigma [25]. Drug use stigma measures included: 1) the Internalized Drug use Stigma Scale [19]; 2) the Substance use stigma mechanism model assessing enacted, anticipated, and internalized stigma [18]; and 3) the Perceived Stigma of Substance Abuse Scale (PSAS) [26].

**Step 2: Forward translation.** Three native Swahili speakers, fluent in English and experienced in conducting translation, independently conducted a forward translation of all the measures from English to Swahili.

**Step 3: Reconciliation.** Reconciliation of the three forward translations was achieved through a panel discussion involving LBM, a psychiatrist experienced in treating PWUD, two people with a history of using drugs, a stigma expert and a linguist proficient in English and Swahili (S1 File).

**Step 4: Back translation.** The reconciled Swahili translation was back translated (S1 File) into English by two new translators who were native speakers of Swahili and fluent in English. The back translators were blinded to the original English source measures.

**Step 5: Back translation review.** LBM reviewed the back translated measures against the English source documents and prepared a discrepancy report highlighting the differences between the two back translations and the original English measures.

**Step 6: Harmonization.** Harmonization was conducted by a panel comprised of a psychiatrist, two people with experience of using drugs, a stigma expert, a linguist proficient in English and Swahili, and LBM. Only the linguist and LBM had participated in Reconciliation. The panel reviewed the original English measures, back translations, and forward translations

**Table 1. Original measures of HIV stigma and drug use stigma that were translated and adapted.**

|   | Measure | Origin | Number of items |
|---|---------|--------|-----------------|
|   | **HIV stigma measures** | | |
| 1 | Abridged Berger Scale [24]<br>• Public attitudes subscale | India | 4 |
| 2 | Internalized AIDS-Related Stigma Scale (IA-RSS) [16] | South Africa | 6 |
| 3 | HIV Stigma Mechanism Measure [25]<br>• Anticipated stigma subscale<br>• Enacted stigma subscale | USA | 18 |
|   | **Drug use stigma measures** | | |
| 1 | Internalized Drug use Stigma Scale [19] | Russia | 6 |
| 2 | Substance use stigma mechanism model [18]<br>• Anticipated stigma subscale<br>• Enacted stigma subscale | USA | 12 |
| 3 | Perceived Stigma of Substance Abuse Scale (PSAS) [26] | USA | 8 |

to make sure the items were translated conceptually and used clear and correct Swahili language that would be understood by PWUD. They addressed all discrepancies through discussion and consensus to develop the final Swahili version of the measures.

**Step 7 (new): Expert review and development of new items.** LBM and a researcher with expertise in stigma who did not participate in earlier steps related to translation, reviewed the content of the drug use stigma and HIV stigma measures. During the expert review, we recognized that across the different measures, there were items which captured similar concepts (Table 1). To avoid duplication, we removed similar items and categorized the remaining items by type of stigma separately for HIV stigma and drug use stigma. We then developed new items reflecting stigmatizing actions known to exist in Tanzania based on the formative research findings by LN, LBM, and JKM (manuscript under review) that were not included in the original measures. We added these new items in all four stigma types (experienced, anticipated, perceived, internalized) for HIV stigma and drug use stigma. In addition, we phrased the new items differently based on the HIV status of PWUD and whether receiving medically assisted treatment or not. That is PWUD living with HIV (S2 File), PWUD not living with HIV (S3 File), people who are on medically assisted treatment and are living with HIV (S4 File) and people who are on medically assisted treatment and are not living with HIV (S5 File). All new items underwent the same process of translation and cultural adaptation that the original items underwent.

**Step 8: Cognitive debriefing.** The final Swahili PROM including finalized and new items was then tested through cognitive debriefing. The aim of the cognitive debriefs was to assess the comprehension of the items of drug use and HIV stigma measures by PWUD.

*Sample size and sampling strategy of participants in the cognitive debriefing.* To assess the comprehension of the items comprising the drug use stigma and HIV stigma measures, we conducted cognitive debriefs (Step 8) with 50 PWUD. We used purposive sampling to recruit participants for the cognitive debriefing, by selecting participants with diverse characteristics including HIV status and MAT status. When conducting a cognitive debrief of measures, it is recommended that 10–15 participants should be debriefed for one item [27]. In this study, each item of the various types of stigma measures was tested among at least 10 participants during the cognitive debriefing. We first conducted cognitive debriefing with 40 PWUD. Out of 40 participants, 19 self-reported they were living with HIV and 21 self-reported a negative or unknown HIV status. Within each group, about half were debriefed using the HIV stigma measures and the other half debriefed with the drug use stigma measures. After reviewing the cognitive debriefing results from the initial 40 participants, we found that some items were not clearly understood. This determination was based on participants' responses where some explicitly said that they did not understand certain words while others interpreted the items differently from their original intent. We therefore revised these items and conducted 10 additional cognitive debriefs among PWUD (half were HIV positive and half were HIV negative or of unknown HIV status) to address the identified issues.

*Cognitive debriefing procedures.* We recruited participants and collected data from April to October 2022. LBM and three trained research assistants (two female, one male) conducted the cognitive debriefing. The research assistants had prior experienced in qualitative data collection and were trained for a week on study objectives, procedures and ethical issues before data collection. We conducted cognitive debriefing in Swahili, either at community-based organization's (CBO) offices or in the community at places frequented by PWUD. The duration of the cognitive interviews ranged from 40 minutes to two hours. We collected data using an interview guide, which consisted of all the items to be adapted. Following each item, four probing questions were asked to assess comprehension. The probing questions were i) Can you tell me in your own words what this question was asking? ii) Did you have any difficulty

using the response options? Why? iii) How did you decide to answer this question? What was your thought process to arrive at your response? iv) Was there a particular experience or experiences that you reflected upon before/while answering the question? For some of the questions where a participant reported that they had experienced stigma, we asked for examples of what happened to him/her. During the initial interviews, we found that participants responded to the third and fourth probes with the same responses although the original intent of the questions were different. Consequently, we dropped the third probe because the fourth probe was more easily understood. The cognitive debriefs were audio recorded and transcribed. Drug use stigma items were asked to all participants. However, for HIV stigma items, participants who were not living with HIV were asked the items related to anticipated and perceived HIV stigma only. Anticipated HIV stigma items were phrased differently for PWUD living with HIV (e.g., How often do you worry that family members will avoid you because you are living with HIV?) and for PWUD not living with HIV (e.g., If you were living with HIV, how often would you be worried that family members would avoid you because you were living with HIV?).

*Data analysis*. At the end of each day of fieldwork, the study team convened to review the data obtained from the cognitive debriefs. We discussed the items, comprehension challenges encountered that day and strategies for addressing them. Following each cognitive debrief, a form was prepared by the research assistants. The feedback was shared with JKM and LN for review and decision. The process was iterative so that whenever we encountered a problem with the item, LBM, JKM and LN discussed that problem to find a solution and those items were adapted and tested in subsequent cognitive debriefs [28]. Authors had no access to information that could identify individual participants during or after data collection.

**Step 9: Review of cognitive debriefing results and finalization.** After data analysis, we reviewed the results of the cognitive debriefing and finalized items and measures.

**Step 10 and 11: Proofreading and final report.** LBM proofread the items and prepared final versions of the adapted measures. We are currently conducting research to determine the adapted measures' psychometric properties and we will report the results separately.

## Ethical consideration

Ethical approval was obtained from the Muhimbili University of Health and Allied Sciences (MUHAS) Senate Research and Publications Committee (MUHAS-REC-06-2021-689) and from the National Health Research Ethics Sub-Committee (NatHREC) of the National Institute for Medical Research, Tanzania (NIMR) (NIMR/HQ/R.8a/Vol.IX/3760). Permission to conduct the study was obtained from Dar es Salaam region authorities and from Northwestern University. Because of the illegality of drug use in Tanzania, participants provided informed verbal consent, an approach used when interviewing a highly stigmatized population engaging in criminalized behavior [29]. We made sure that confidentiality and privacy were maintained throughout the study.

## Results

Fifty participants participated in the cognitive debriefing. Among those, 52% were living with HIV, 78% were male, 56% were on MAT, and 56% were currently using drugs. Most of the participants (48%) were aged 36 to 45 years; only one participant (2%) was between 18–24 years, and three participants (6%) were older than 55 years (Table 2).

### Steps 1–6: Related to translation

**English words that could be translated into more than one Swahili word.** All 54 original items and newly developed items covering 12 stigmatizing actions were translated into Swahili.

**Table 2. Demographic characteristics of participants.**

| Variable | n | % |
|---|---|---|
| **MAT** | | |
| On MA | 22 | 44 |
| Not on MAT | 28 | 56 |
| **Sex** | | |
| Male | 39 | 78 |
| Female | 11 | 22 |
| **Drug use** | | |
| Current use | 28 | 56 |
| Past use | 22 | 44 |
| **HIV status** | | |
| Positive | 26 | 52 |
| Negative | 24 | 48 |
| **Age** | | |
| 18–24 | 1 | 2 |
| 25–35 | 14 | 28 |
| 36–45 | 24 | 48 |
| 46–55 | 8 | 16 |
| >55 | 3 | 6 |

Some English words capturing key dimensions of stigma had multiple possible translations into Swahili. For example: "avoided me" could be translated as "wamenikwepa" or "wamenie-puka". "Treated me differently" could be translated as "watanichukulia tofauti" or "wataniten-dea tofauti". During reconciliation and harmonization, for all the items with multiple possible translation options, the panels reached a consensus to select the word that best fit in the context in which it would be used, that is the word that PWUD would understand easily and maintained the original intent of the item.

## Step 7: Expert review of original measures and development of new items

Table 3 shows the content of the original measures and new items drafted to capture important missing concepts. Examples of new items are in Tables 4 and 5.

**Removed items related to uncommon types of health care providers interacting with PWUD.**   For the experienced HIV stigma items and anticipated HIV stigma items from the HIV Stigma Mechanism Measure [25] (Tables 4 and 5), we adapted six of the nine items in both categories of stigma because the remaining three items were about stigma actions PWUD experience from community or social workers whom PWUD do not interact with as much as they do with health care workers.

**Changed the response options.**   For the experienced HIV stigma items from the HIV Stigma Mechanism Measure [25], we changed the response options from "never", "not often", "somewhat often", "often", and "very often" to "never", "once", "a few times", "often" because of translation challenge with the phrase "somewhat often" and participants had a clearer understanding of the response options never, once, a few times, and often as more concrete frequencies (Tables 4 and 5). This change was made for anticipated HIV stigma items from the HIV Stigma Mechanism Measure, and experienced and anticipated drug use stigma from the Substance use stigma mechanism model.

**Table 3. Verbs of original and new items of the measures of HIV stigma and drug use stigma.**

| Verbs of Original items | Verbs of New items |
|---|---|
| Cannot be trusted | Judged |
| Looked down on me | Shamed |
| Treated me differently | Blame/blamed |
| Not listened to my concerns | Ignored |
| Pill shopping | Talked badly or gossip |
| Given me poor care | Verbally harass |
| Accept someone | Disclose HIV and drug use status |
| Being teacher of young children | Steal |
| Hired to care for their child | Wait longer to receive care than other clients |
| Think less of | Avoid physical contact |
| Hire if qualified | Denied care if not dressed properly |
| Pass over application | Given priority and treated first instead of other clients |
| Dating | |
| Being rejected | |
| Losing jobs | |
| Disgusting | |
| Treat with less respect | |

All items of the Internalized drug use stigma measure [19] were negatively phrased. Its response options were "strongly agree," "agree," "neither agree nor disagree," "disagree," and "strongly disagree". We reversed the order of the response options to be in a direction similar to that of the other stigma measures. We adapted internalized HIV stigma items from IA-RSS [16]. In the original measure, the response options were "agree" and "disagree". We added more response options in order to be similar with the Internalized drug use stigma measure. The revised response options were "strongly disagree", "disagree", "neither disagree nor agree", "agree, strongly agree". The same change was made for the response options of the Perceived Stigma of Substance Abuse Scale (PSAS) items from the response options of "very strongly disagree, strongly disagree, disagree, agree, strongly agree, and very strongly agree" to "strongly disagree, disagree, neither disagree nor agree, agree, strongly agree" (Table 6).

**Changing the phrasing of selected questions.** For the anticipated HIV stigma items from the HIV Stigma Mechanism Measure, we changed the questions from "How likely is it that people will treat you in the following ways in the future because of your HIV status?" to "How often do you worry that family members will (example avoid) you because you are living with HIV?" in order to assess the frequency of stigma and to be similar to the other items in the experienced HIV stigma measure (Table 4). We culturally adapted six items related to experienced drug use stigma from the Substance use stigma mechanism model. We changed the original items from a statement to a question format in order to assess the frequency of experiences of stigma, have a parallel style with other adapted measures, and use response options which are more concrete and easier to translate as occurred with the other types of stigma (Table 5). We culturally adapted eight perceived drug use stigma items from the Perceived Stigma of Substance Abuse Scale (PSAS). We changed the original items' focus on issues related to substance use in general to issues related to drug use. The term drug use is specific for illicit drugs while substance use is a general term referring to various substances including alcohol (Table 6).

**Changed wording of questions based on culture value placed on dating and marriage.** For the items of the Perceived Stigma of Substance Abuse Scale (PSAS), we changed the eighth item: "Most people would be willing to *date* someone who has been treated for substance use"

**Table 4. Anticipated HIV stigma measures with original and culturally adapted items.**

| | Original anticipated stigma items | | Culturally adapted anticipated stigma items |
|---|---|---|---|
| Introduction | **How likely is it that people will treat you in the following ways in the future because of your HIV status?** | | - |
| Response options | **Very unlikely, Unlikely, Neither unlikely nor likely, Likely Very likely** | | **Never, Once, A few times, Often** |
| 1 | Family members will avoid me | 1 | How often do you worry that family members will avoid you because you are living with HIV? |
| 2 | Family members will look down on me | 2 | How often do you worry that family members will look down on you because you are living with HIV? |
| 3 | Family members will treat me differently | 3 | How often do you worry that family members will treat you differently because you are living with HIV? |
| 4 | Community/social workers won't take my needs seriously | | Not adapted |
| 5 | Community/social workers will discriminate against me | | Not adapted |
| 6 | Community/social workers will deny me services | | Not adapted |
| 7 | Healthcare workers will not listen to my concerns | 4 | How often do you worry that healthcare workers will not listen to your concerns when you go to a health facility for general health care because you are living with HIV? |
| 8 | Healthcare workers will avoid touching me | 5 | How often do you worry that healthcare workers will avoid physical contact with you when you go to a health facility for general health care because you are living with HIV? |
| 9 | Healthcare workers will treat me with less respect | 6 | How often do you worry you will be disrespected when you go to a health facility for general health care (, because you are living with HIV? |
| | | 7 | How often do you worry that family members will think that you cannot be trusted because you are living with HIV? |
| | | 8 | How often do you worry that your peers/other people who use or used drugs will avoid you because you are living with HIV? |
| | | 9 | How often do you worry your peers/other people who use or used drugs will gossip about you because you are living with HIV? |
| | | 10 | How often do you worry that your peers/other people who use/used drugs will reject you because you are living with HIV? |
| | | 11 | How often do you worry that people who know you have HIV will tell others? |
| | | 12 | How often do you worry that people may judge you when they learn you have HIV? |
| | | 13 | How often do you worry that healthcare workers will give you poor care when you go to a health facility for general health care because you are living with HIV? |
| | | 14 | How often do you worry that you will be made to wait longer than other clients when you go to a health facility for general health care because you are living with HIV? |
| | | 15 | How often do you worry that you will be judged when you go to a health facility for general health care because you are living with HIV? |
| | | 16 | How often do you worry that you will be shamed when you go to a health facility for general health care because you are living with HIV? |
| | | 17 | How often do you worry that you will be blamed when you go to a health facility for general health care because you are living with HIV? |
| | | 18 | How often do you worry that you will be ignored when you go to a health facility for general health care (because you are living with HIV? |
| | | 19 | How often do you worry that health care workers will disclose to others that you are living with HIV when you go to a health facility for general health care? |
| | | 20 | How often do you worry that you will be verbally harassed (e.g., yelled ag, scolded) when you go to the health facility for general health care, because you are living with HIV |

**Table 5. Experienced HIV stigma measures with original and culturally adapted items.**

| | Original experienced stigma items | | Culturally adapted experienced stigma items |
|---|---|---|---|
| Introduction | **How often have people treated you this way in the past because of your HIV status?** | | - |
| Response options | **Never, Not often, Somewhat often, Often, Very often** | | **Never, Once, A few times, Often** |
| 1 | Family members have avoided me | 1 | How often have family members avoided you because you are living with HIV |
| 2 | Family members have looked down on me | 2 | How often have family members looked down on you because you are living with HIV |
| 3 | Family members have treated me differently | 3 | How often have family members treated you differently because you are living with HIV |
| 4 | Community /social workers have not taken my needs seriously | | Not adapted |
| 5 | Community /social workers have discriminated against me | | Not adapted |
| 6 | Community/social workers have denied me services | | Not adapted |
| 7 | Healthcare workers have not listened to my concerns | 4 | How often have healthcare workers not listened to your concerns when you have gone to a health facility for general care because you are living with HIV? |
| 8 | Healthcare workers have avoided touching me | 5 | How often have healthcare workers avoided physical contact with you when you have gone to a health facility for general health care, because you are living with HIV? |
| 9 | Healthcare workers have treated me with less respect | 6 | How often have healthcare workers treated you disrespectfully when you have gone to a health facility for general health care (because you are living with HIV? |
| | | 7 | How often have family members talked badly or gossiped about you because of your HIV status? |
| | | 8 | How often have other people (not family members) talked badly gossiped about you because of your HIV status? |
| | | 9 | How often has someone verbally harassed you (e.g., yell, scold) because of your HIV status? |
| | | 10 | How often have peers (people who use or used drugs) talked badly or gossiped about you because of your HIV status? |
| | | 11 | How often have healthcare workers verbally harassed you (e.g., yelled, scolded) when you have gone to a health facility for general health care because you are living with HIV? |

to "Most people would be willing to *marry* someone who has been treated for drug use" as dating in the Tanzanian cultural context is not common (Table 6).

## Step 8 and 9: Cognitive debriefing and review of results

**Challenges with interpretation.** In general, most of the original items (42/54, 78%) and response options and all items on the 12 new stigmatizing actions were understood as intended by the participants. Only a few items were not understood as intended by all participants. These included two items in the "Experienced HIV stigma" measure, (Item 3: Family members have treated me differently and Item 9: Healthcare workers have treated me with less respect); two items in the "Anticipated HIV stigma" measure which have the same stems as those in the experienced HIV stigma measure (Item 3: Family members will treat me differently and Item 9: Healthcare workers will treat me with less respect); one item in the "Perceived HIV stigma" measure (Item 1: Most people believe that a person who has HIV is dirty) and one item in the "Internalized HIV stigma" measure (2: Being HIV positive makes me feel dirty).

For drug use stigma measures, items that were not understood as intended by all participants included: one item in the "Experienced drug use stigma" measure (Item 3: Family members have treated me differently) and one item in the "Anticipated drug use stigma" measure which also has the same stem as the one in the experienced drug use stigma measure (Family members will treat me differently); two items in the "Perceived drug use stigma" measure

**Table 6. Perceived drug use stigma measures with original and culturally adapted items.**

| | Original items | | Culturally adapted items |
|---|---|---|---|
| Response options | Very strongly disagree, Strongly disagree, Disagree, Agree, Strongly agree, and Very strongly agree | | Strongly disagree, Disagree, Neither disagree nor agree, Agree, Strongly agree |
| 1 | Most people would willingly accept someone who has been treated for substance use as a close friend | 1 | Most people would willingly accept someone who has been treated for drug use as a close friend (not a sexual partner) |
| 2 | Most people believe that someone who has been treated for substance use is just as trustworthy as the average citizen | 2 | Most people believe that someone who has been treated for drug use is just as trustworthy as the average citizen |
| 3 | Most people would accept someone who has been treated for substance use as a teacher of young children in a public school | 3 | Most people would accept someone who has been treated for drug use as a teacher of children in a primary school |
| 4 | Most people would hire someone who has been treated for substance use to take care of their children | 4 | Most people will hire someone who has been treated for drug use to take care of their children |
| 5 | Most people think less of a person who has been in treatment for substance use | 5 | Most people think less of a person who has been in treatment for drug use |
| 6 | Most employers will hire someone who has been treated for substance use if he or she is qualified for the job | 6 | Most employers will hire someone who has been treated for drug use if he or she is qualified for the job |
| 7 | Most employers will pass over the application of someone who has been treated for substance use in favor of another applicant | 7 | Most employers will pass over the application of someone who has been treated for drug use in favor of another applicant |
| 8 | Most people would be willing to date someone who has been treated for substance use | 8 | Most people would be willing to marry someone who has been treated for drug use |
| | | 9 | Most people believe that a person who uses drugs is dirty |
| | | 10 | Most people who use drugs are rejected when others find out that they use drugs |
| | | 11 | People who use drugs lose their jobs when their employers find out they use drugs |
| | | 12 | Most people think that a person who uses drugs is disgusting |
| | | 13 | Most people judge people who use drugs |
| | | 14 | Most people do not respect people who use drugs |
| | | 15 | Most people think people who use drugs have only themselves to blame for their drug use |
| | | 16 | Most people verbally harass (e.g., yell, scold) people who use drugs |

(Item 1: Most people would willingly accept someone who has been treated for substance use as a close friend, Item 3: Most people would accept someone who has been treated for substance use as a teacher of young children in a public school); and one item in the "Internalized drug use stigma" measure (Item 2: Using drugs makes me feel dirty).

In the item "family members will treat me differently", the words *"treat me differently"* which in Swahili means *"kutendewa kwa namna tofauti"*, was not understood by one participant, who queried as quoted below:

"Interviewer: how often have family members treated you differently because you are living with HIV?

Respondent: what does it mean to be treated?"

All of the other participants explained the meaning of "being treated differently" by giving examples of stigmatizing actions such as being avoided, gossiped about by others. After testing the item in the additional ten interviews, some participants provided the correct meaning of the words and others gave examples of stigmatizing actions to explain the meaning indicating that they understood the terminology. We concluded that the words "treated differently" are understood and decided to keep them as they were.

In the question *"have healthcare workers treated you with less respect when you have gone to a health facility for general health care because you are living with HIV?"*, the problem was with the words *"less respect"* which translates to *"heshima ndogo"* in Swahili which does not make much sense, as linguistically in Swahili, respect is either present or absent and is not often qualified in levels. We therefore decided to change the words *"less respect"* to *"disrespect"* which translated to *"bila heshima"* (English without respect) in Swahili for clarity.

"Participant: Aaaah..the way I have understood the question, it is less respect.

Interviewer: Okay, by less respect, how?

Participant: I mean they did not respect me because of drug use."

A similar challenge was found in the item *"Most people would willingly accept someone who has been treated for substance use as a close friend"*. The problem was on the interpretation of the words *"close friend"* which translates to *"rafiki wa karibu"* in Swahili. Some participants interpreted "a close friend" as "a sexual partner".

"Interviewer: What do you understand by the words close friend?

Participant: A close friend has two meanings, a sexual partner or a normal friend."

We therefore, used the words "close friend", and added the words "not a sexual partner" which translates to *"sio rafiki wa kimapenzi"* (English: "not a sexual partner") to clarify that a close friend was not meant to include a sexual partner. Hence the item read *""Most people would willingly accept someone who has been treated for substance use as a close friend (not a sexual partner)"*.

Another challenge we encountered was the difference in the interpretation of words in specific items among participants. In the item *"most people would accept someone who has been treated for substance use as a teacher of young children in a public school"*, we found out that, the words *"young children"* and *"public school"* were interpreted differently among participants. The words "young children" which translate to *"watoto wadogo"* in Swahili, were interpreted by participants as either children in kindergarten or children in primary school or all children below 18 years. We therefore revised those words to "children in primary school" as it was the closest meaning of the item. In addition, the words *"public school"* in Swahili mean *"shule ya umma"*. Some participants understood those words as schools owned by the government while other participants understood them as schools owned by both government and private individuals. We therefore decided to remove the word public and leave the word schools only to eliminate this problem of different interpretation.

"Interviewer: what does public schools mean?

Participant: Public schools mean government schools."

The other items with a problem with interpretation had the word 'dirty'. For example, item: "Using drugs makes me feel dirty", in this item, the word dirty was interpreted differently among participants, some interpreted it as being dirty physically such as not taking a bath, or wearing dirty clothes, while others interpreted it as being someone who has made a mistake.

We decided to leave it as it is and wanted to decide whether it fits in the measure during psychometric testing.

## Step 10 and 11: Proofreading and final report

LBM proofread the items and prepared final versions of adapted measures. The adapted measures—drug use stigma and HIV stigma—consisted of items assessing experienced, anticipated, perceived and internalized stigma.

## Discussion

In this study, we have described the process of translating and culturally adapting both HIV stigma and drug use stigma measures among PWUD in Tanzania. We used Wild's method and modified it by adding one step of reviewing stigma measures before cognitive debriefing. All steps in Wild's method were useful during the translation and adaptation process. The added step was also useful because it enabled us to ensure a variety of stigmatizing actions reflecting the Tanzanian context were captured in the measures. Adding a new step in the translation and adaptation of measures is common [30]. After translation and adaptation, most of the of the items were understood and relevant to the participants, although some challenges were identified which required adaptation of items, wording, and response options. These results are similar and consistent with the literature around linguistic translations and cultural adaptation [31–34].

Translating items of measures from the original language can result in having alternatives of words to use in the target language [31, 35]. Our results are similar to this observation where during the translation, we encountered the challenge of having one English word which could be translated in multiple choices of words in Swahili. We therefore initially selected the wording based on feedback from the translation and reconciliation panel meetings held during the process of translating items to be adapted and later confirmed the wording based on the recommendations of the participants in the cognitive debriefing. It is therefore necessary to have a variety of key individuals—experts and people with lived experience of the condition—involved in the translation process to ensure that the most preferred translated word is chosen to use in the final measure. In addition, we encountered differences between participants in the interpretation of the phrases or words necessitating changes that focused on making specific what was being asked. A similar challenge in the interpretation of the word "children" for example is reported in a study involving caregivers participants in Jordan [36].

Measures can be developed by reviewing original measures adapted from various researchers [37]. Reviewing original measures is an essential step to enable identification of missing or redundant concepts in the items to be adapted. In our study, after reviewing HIV stigma and drug use stigma items, we found that some stigmatizing actions reflecting the Tanzanian context were not included in the original measures that we wanted to adapt. As we needed to have all the possible stigmatizing actions captured in the measures, we added new stigmatizing actions in the four types of stigma for both measures of drug use stigma and HIV stigma. In addition, during the review of the stigma items, we removed some items about the types of providers the Tanzanian PWUD encounters. In Tanzania, PWUD typically do not interact with community or social workers; we hence dropped items referring to these providers. A similar challenge of measures referencing health care worker not commonly used in a community was observed when adapting a measure in Afrikaans in South Africa [31].

During translation and cultural adaptation of measures, items can be revised as a result of the cultural differences between the original and target language [31, 38]. The revisions are made using the words that are understood and fit the context of the target language and

population. In our study, we found cultural differences with some behaviors described in the items that needed revisions to reflect the Tanzanian context. The behavior of dating for example is not very common in Tanzania. We therefore, used marriage as a close approximate, as in some Tanzanian communities, particularly where socio-economic status is low, women marry at a young age without necessarily dating in part due to the financial benefit of a dowry paid to the bride's family [39].

We also revised the response options of the items of the stigma measures. Revision of response options is common when adapting measures. Adjusting response options allows fitting the language to the context of the target population and ensures clarity during interviews. This process of revising response options during adaptation has also been reported in South Africa [40].

Experiences of participants not understanding the meaning of some items during translation and adaptation of measures is common [32, 40]. Our study showed similar findings where some participants did not understand some words used in the items. Failure of participants to understand some items is common and understandable because the measures were developed in foreign countries with different contexts and cultures. As language is one of the components of culture, it is expected that it may differ from one context to another. Therefore, based on the results of this study, we learned that it is crucial to conduct both translation and cultural adaptation of measures in order to address the differences in language that may be present between the original language and the language of the target population that will use the measure.

## Strengths and limitations

Our study has some strengths. First, our study included a sufficient number of participants during the cognitive debriefing. It is suggested that when adapting measures, 10–15 interviews are to be conducted for one item [27]. We had many items due to the four categories of drug use and HIV stigma, namely experienced, anticipated, perceived and internalized. In our study, at least ten participants responded to the items in a specific category. Second, the translation process involved a variety of stakeholders who are key in the process of translation and adaptation of measures among people who use drugs. These included psychiatrists, stigma experts, people who had experience of using drug, and a linguist. Third, we have developed new items covering stigmatizing actions common in the Tanzanian context that were not captured in the original measures. Similar research has been conducted where measures were translated and culturally adapted, and were available for practice and further validation [30]. Despite these strengths, our study has one limitation. We did not adapt all the items in the HIV stigma mechanism measure; we removed three items for the anticipated HIV stigma subscale and three items for the experienced HIV stigma subscale. This might affect the validity of the adapted measures.

## Conclusion

HIV stigma and drug use stigma measures can successfully be translated and culturally adapted among PWUD in Tanzania. Many of the items were understood by participants and were found to be relevant to the population of people who use drugs. We are currently conducting research to determine the stigma measures' psychometric properties and we will report the results separately. This work is important as there were no such measures available which had been linguistically and culturally adapted for use among people who use drugs in Tanzania and which measured across the four domains of stigma. The availability of these

measures is important because they will enable key stakeholders to measure stigma among PWUD, design interventions and measure for effectiveness.

## Supporting information

**S1 File. Forward and back translated HIV and drug use stigma measures.**
(DOCX)

**S2 File. Stigma measures for people who use drugs who are living with HIV.**
(DOCX)

**S3 File. Stigma measures for people who use drugs who are not living with HIV.**
(DOCX)

**S4 File. Stigma measures for people who are on medically assisted treatment who are living with HIV.**
(DOCX)

**S5 File. Stigma measures for people who are on medically assisted treatment who are not living with HIV.**
(DOCX)

## Acknowledgments

We would like to thank everyone who participated in the translation and cultural adaptation of the stigma tools including translators, psychiatrists, a linguist, people with history of drug use who participated in the panel and the CI participants. We also acknowledge the support from research assistants and the participants of this study for their valuable contribution towards the success of this study.

## Author Contributions

**Conceptualization:** Linda B. Mlunde, Lisa R. Hirschhorn, Laura Nyblade, Judith T. Moskowitz, Sylvia Kaaya, Germana Leyna, Jessie K. Mbwambo.

**Formal analysis:** Linda B. Mlunde, Jessie K. Mbwambo.

**Methodology:** Linda B. Mlunde, Lisa R. Hirschhorn, Laura Nyblade, Nan E. Rothrock, Erasto V. Mbugi, Judith T. Moskowitz, Sylvia Kaaya, Claudia Hawkins, Germana Leyna, Jessie K. Mbwambo.

**Project administration:** Linda B. Mlunde.

**Validation:** Linda B. Mlunde.

**Writing – original draft:** Linda B. Mlunde.

**Writing – review & editing:** Linda B. Mlunde, Lisa R. Hirschhorn, Laura Nyblade, Nan E. Rothrock, Erasto V. Mbugi, Judith T. Moskowitz, Sylvia Kaaya, Claudia Hawkins, Germana Leyna, Jessie K. Mbwambo.

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
