## [Decision Letter · Decision Letter 0]

22 Aug 2023

PONE-D-23-13168Translation and cultural adaptation of drug use stigma and HIV stigma measures among people who use drugs in TanzaniaPLOS ONE

Dear Dr. Mlunde,

Thank you for submitting your manuscript to PLOS ONE. After careful consideration, we feel that it has merit but does not fully meet PLOS ONE’s publication criteria as it currently stands. Therefore, we invite you to submit a revised version of the manuscript that addresses the points raised during the review process. Please submit your revised manuscript by Oct 06 2023 11:59PM. If you will need more time than this to complete your revisions, please reply to this message or contact the journal office at plosone@plos.org. Please include the following items when submitting your revised manuscript:A rebuttal letter that responds to each point raised by the academic editor and reviewer(s). You should upload this letter as a separate file labeled 'Response to Reviewers'.A marked-up copy of your manuscript that highlights changes made to the original version. You should upload this as a separate file labeled 'Revised Manuscript with Track Changes'.An unmarked version of your revised paper without tracked changes. You should upload this as a separate file labeled 'Manuscript'.If applicable, we recommend that you deposit your laboratory protocols in protocols.io to enhance the reproducibility of your results. Protocols.io assigns your protocol its own identifier (DOI) so that it can be cited independently in the future. For instructions see: https://journals.plos.org/plosone/s/submission-guidelines#loc-laboratory-protocols. Additionally, PLOS ONE offers an option for publishing peer-reviewed Lab Protocol articles, which describe protocols hosted on protocols.io. Read more information on sharing protocols at https://plos.org/protocols?utm_medium=editorial-email&utm_source=authorletters&utm_campaign=protocols.

We look forward to receiving your revised manuscript.

Kind regards,

Alexander Röhm

Guest Editor

PLOS ONE

Journal Requirements:

"..This research was supported by the Fogarty International Center of the National Institute of Health (NIH), Award Number D43TW010946. The research content is solely the responsibility of the authors, and the NIH had no role in study design, data collection and analysis, decision to publish or preparation of the manuscript."

"The study was supported by the Fogarty International Center of the National Institutes of Health (NIH) under Award Number D43TW010946. The research is solely the responsibility of the authors, and the NIH had no role in study design, data collection and analysis, decision to publish, or preparation of the manuscript. No any, among authors have received salary from NIH."

**Additional Editor Comments:**

I thank the authors for this very important and well written study. I agree with the reviewers' general evaluation that the manuscript requieres only minor revision before it is fit for publication. However, I recommend to pay close attention to the following issues, which may have also been pointed out by the reviewers:

1) Please state more clearly that this study does not yet include a psychometric evaluation of the scale (see also comment from reviewer 2), which is currently only mentioned at the end of the manuscript.

2) I really appreciate that the authors fully disclose all documents used in their study. However, concerning the strict data sharing policy of PLOS, I suggest to also share information on the translation of each item (original item -> translated item -> back translation) as well as the (english translated) transcripts from the interviews, since the cited responses and, thus, the qualitative evaluation strongly depend on that data.

3) Please proofread the whole manuscript after revision to improve language quality as also suggested by reviewer 1.

I am looking forward to recieving your revised manuscript.

Reviewers' comments:

Reviewer's Responses to Questions

**Comments to the Author**

1. Is the manuscript technically sound, and do the data support the conclusions?

Reviewer #1: Yes

Reviewer #2: Yes

2. Has the statistical analysis been performed appropriately and rigorously? 

Reviewer #1: Yes

Reviewer #2: N/A

3. Have the authors made all data underlying the findings in their manuscript fully available?

Reviewer #1: Yes

Reviewer #2: Yes

4. Is the manuscript presented in an intelligible fashion and written in standard English?

Reviewer #1: No

Reviewer #2: Yes

5. Review Comments to the Author

Reviewer #1: The study presents the results of the original work. The steps of translation and adaptation of the scales has been explained in detailed form which makes it easy for the reader to understand the process.

Important note: The study reports on the process of translating and adapting the scales and does not report the psychometric properties of the scales. When reporting the psychometric properties, it is recommended that each scale is discuss in a separate article as this will enable clear communication of the results.

Method: The study follows Wild's 10 steps. It is recommended that the explanation of the steps and the details of the study (how the steps are used in the study) are reported together instead of discussing them separately as it leads to unnecessary confusions for the reader. For example, Line 220-237 Sample size and sampling strategy of participants in the cognitive debriefing should be added with the explanation of step 8- Cognitive debriefing (line 196-199).

Study design: Line 136- Authors have reported formative assessment, it is recommended that the mentioned formative assessment be discussed briefly and cited.

Six scales have been reported to be translated, it is recommended to add their brief overview

Proofreading is recommended to clear a few language and typing errors

Abbreviations mentioned should be written in full form the first time they are mentioned in the manuscript.

Reviewer #2: There are a few typos through out, I would proofread this paper.

I commend the authors on the major task of 50 cognitive interviews, this far surpasses standard procedure and literature recommendation. I think that it is a shame that the authors do not embark on psychometric analysis on a new sample, as that would significantly enhance the paper.

6. PLOS authors have the option to publish the peer review history of their article (what does this mean?). If published, this will include your full peer review and any attached files.

Reviewer #1: No

Reviewer #2: No

---

## [Author Response · Author response to Decision Letter 0]

20 Sep 2023

Responses to Academic Editor comments

Response: 

Thank you very much. We have ensured that our manuscript meets PLOS ONE’s style requirements including headings and supporting information.

"..This research was supported by the Fogarty International Center of the National Institute of Health (NIH), Award Number D43TW010946. The research content is solely the authors' responsibility, and the NIH had no role in study design, data collection, and analysis, decision to publish or preparation of the manuscript."

"The study was supported by the Fogarty International Center of the National Institutes of Health (NIH) under Award Number D43TW010946. The research is solely the responsibility of the authors, and the NIH had no role in study design, data collection and analysis, decision to publish, or preparation of the manuscript. No any, among authors have received salary from NIH."

Please include your amended statements within your cover letter; we will change the online submission form on your behalf. We have removed the funding statement from the acknowledgments section. 

Response: 

We have added information in the Acknowledgements section about research assistants and participants.

We have included our Funding statement within the cover letter. Please see this on page 30, Line 564 – 566.

3 Please include captions for your Supporting Information files at the end of your manuscript, and update any in-text citations to match accordingly. Please see our Supporting Information guidelines for more information: http://journals.plos.org/plosone/s/supporting-information. 

Response

Thank you for the comment. We have added captions of Supporting Information files at the end of the manuscript and updated the in-text citations to match accordingly. Please see this on page 9, Line 173, 176; on page 10, Line 206 – 208; and on page 33, Line 691 – 697.

4 Please review your reference list to ensure that it is complete and correct. If you have cited papers that have been retracted, please include the rationale for doing so in the manuscript text, or remove these references and replace them with relevant current references. Any changes to the reference list should be mentioned in the rebuttal letter that accompanies your revised manuscript. If you need to cite a retracted article, indicate the article’s retracted status in the References list and also include a citation and full reference for the retraction notice. 

Response

Thank you. The references have been reviewed and revised where there were errors. Please see the revisions on page 30, Line 569 – 570, 575, 578 – 581; page 31, Line 589 – 594; and on page 32, Line 633 – 635, 645 – 650, 661 – 662.

5. I thank the authors for this very important and well written study. I agree with the reviewers' general evaluation that the manuscript requieres only minor revision before it is fit for publication. However, I recommend to pay close attention to the following issues, which may have also been pointed out by the reviewers:

1) Please state more clearly that this study does not yet include a psychometric evaluation of the scale (see also comment from reviewer 2), which is currently only mentioned at the end of the manuscript. Thank you. 

Response

We are researching the stigma measures’ psychometric properties, and we will report the results separately. We have added this information to the manuscript. Please see this in the Abstract on page 3, Lines 56 -58, in the Introduction on page 6, Lines 111 – 113, and in the main text on page 13, Line 276 – 278; and on page 30, Lines 555 – 556.

6 I really appreciate that the authors fully disclose all documents used in their study. However, concerning the strict data sharing policy of PLOS, I suggest to also share information on the translation of each item (original item -> translated item -> back translation) as well as the (english translated) transcripts from the interviews, since the cited responses and, thus, the qualitative evaluation strongly depend on that data. 

Response

Thank you. The information on the translation process is in the supplement (S1 File). We cannot share the participant transcripts because we did not ask for participants’ consent to share narrative data with an external third party. Upon a reasonable request, we can ask for permission from the IRBs for an exemption (contact Dr. Mlunde).

7 Please proofread the whole manuscript after revision to improve language quality as also suggested by reviewer 1. Response

Thank you. We have done this.

Responses to reviewers' comments

Reviewer #1:

1 The study presents the results of the original work. The steps of translation and adaptation of the scales have been explained in detail, making it easy for the reader to understand the process. 

Response

Thank you very much for your comment.

2 Important note: The study reports on the process of translating and adapting the scales and does not report the psychometric properties of the scales. When reporting the psychometric properties, it is recommended that each scale is discuss in a separate article as this will enable clear communication of the results. 

Response

Thank you. We are researching the stigma measures’ psychometric properties, and we will report the results separately. We have added this information to the manuscript.

3 Method: The study follows Wild's 10 steps. It is recommended that the explanation of the steps and the details of the study (how the steps are used in the study) are reported together instead of discussing them separately as it leads to unnecessary confusions for the reader. 

For example, Line 220-237 Sample size and sampling strategy of participants in the cognitive debriefing should be added with the explanation of step 8- Cognitive debriefing (line 196-199). 

Response

The steps have been reported together. We have combined all the details related to Step 8. 

4 Study design: Line 136- Authors have reported formative assessment, it is recommended that the mentioned formative assessment be discussed briefly and cited. 

Response

We have added the below text in the manuscript:

“The formative assessment was qualitative in design where it included in-depth interviews of 18 PWUD and 14 HIV care and treatment clinics (CTC) staff. The aim of the formative assessment was to understand the types of stigma and their manifestations that PWUD face.“

5 Six scales have been reported to be translated, it is recommended to add their brief overview 

Response

The brief overview is available on the table originally submitted. The information included was on measures’ types of stigma, number of items, and country of origin added.

6 Proofreading is recommended to clear a few language and typing errors 

Response

We have done proofreading to correct these errors.

7 Abbreviations mentioned should be written in full form the first time they are mentioned in the manuscript. 

Response

We have corrected this.

Reviewer #2: 

1 There are a few typos through out, I would proofread this paper. 

Response

We have done proofreading to correct typos and improve clarity. 

2 I commend the authors on the major task of 50 cognitive interviews. This far surpasses standard procedure and literature recommendations. It is a shame that the authors do not conduct psychometric analysis on a new sample, which would significantly enhance the paper. 

Response

Thank you very much.

We are researching the stigma measures’ psychometric properties, and we will report the results separately. We have added this information to the manuscript. Please see this in the Abstract on page 3, Lines 56 -58, in the Introduction on page 6, Lines 111 – 113, and in the main text on page 13, Line 276 – 278; and on page 30, Lines 555 – 556.

---

## [Editor Report · Decision Letter 1]

26 Sep 2023

Translation and cultural adaptation of drug use stigma and HIV stigma measures among people who use drugs in Tanzania

PONE-D-23-13168R1

Dear Dr. Mlunde,

We’re pleased to inform you that your manuscript has been judged scientifically suitable for publication and will be formally accepted for publication once it meets all outstanding technical requirements.

Kind regards,

Alexander Röhm

Guest Editor

PLOS ONE

Additional Editor Comments (optional):

Thank you for your timely revision of the manuscript. I am very happy to inform you that it can be accepted in the current form. Since I could not find any statement regarding the restrictions, but potential availability of the interview data upon request, please update or include such statement into your final version.
---

## [Editor Report · Acceptance letter]

10 Oct 2023

PONE-D-23-13168R1 

Translation and cultural adaptation of drug use stigma and HIV stigma measures among people who use drugs in Tanzania 

Dear Dr. Mlunde:

I'm pleased to inform you that your manuscript has been deemed suitable for publication in PLOS ONE. Congratulations! Your manuscript is now with our production department. 

Kind regards, 

on behalf of

Dr. Alexander Röhm 

Guest Editor

PLOS ONE